# Systematic Investigation of Aluminum Stress-Related Genes and Their Critical Roles in Plants

**DOI:** 10.3390/ijms25169045

**Published:** 2024-08-21

**Authors:** Chaowei Fang, Jiajing Wu, Weihong Liang

**Affiliations:** 1College of Life Science, Henan Normal University, Xinxiang 453007, China; fangchaowei@htu.edu.cn; 2Xinxiang Academy of Agricultural Sciences, Xinxiang 453000, China; wujj109@163.com

**Keywords:** Al stress, Al stress response gene, alleviating Al toxicity, crop production, plants

## Abstract

Aluminum (Al) stress is a dominant obstacle for plant growth in acidic soil, which accounts for approximately 40–50% of the world’s potential arable land. The identification and characterization of Al stress response (Al-SR) genes in *Arabidopsis*, rice, and other plants have deepened our understanding of Al’s molecular mechanisms. However, as a crop sensitive to acidic soil, only eight Al-SR genes have been identified and functionally characterized in maize. In this review, we summarize the Al-SR genes in plants, including their classifications, subcellular localizations, expression organs, functions, and primarily molecular regulatory networks. Moreover, we predict 166 putative Al-SR genes in maize based on orthologue analyses, facilitating a comprehensive understanding of the impact of Al stress on maize growth and development. Finally, we highlight the potential applications of alleviating Al toxicity in crop production. This review deepens our understanding of the Al response in plants and provides a blueprint for alleviating Al toxicity in crop production.

## 1. Introduction

Acidic soil is globally widespread, encompassing approximately 40–50% of the world’s potentially arable lands, and it constrains crop production worldwide significantly [1,2]. As the most abundant metal element in the earth’s crust, aluminum (Al) mainly exists as insoluble aluminosilicates or Al oxides, which are non-toxic to plant growth, while it exhibits high toxicity toward plants of Al^3+^ in acidic environments (pH < 5.5) [3]. The predominant obstacle to plant growth in acidic soil is commonly attributed to Al toxicity [4]. Thus, the exploration of the toxic mechanism of Al stress and the characterization of the Al stress response (Al-SR) genes in plants will facilitate potential applications for alleviating Al stress, as well as the crop breeding of and genetic improvement in Al-tolerant varieties.

The effects of Al toxicity on plants are irreversible, even in the presence of a micromolar concentration of Al in the soil [4]. Al toxicity is associated with the interaction between Al and the cell walls, plasma membranes, and symplasms of apical root cells in plants [5]. The primary manifestation of Al stress on plants is the suppression of root elongation, subsequently leading to the restricted uptake of water and nutrients [6,7]. For self-protection, plants have evolved strategies to cope with Al stress, among which internal tolerance and external exclusion are widely considered the primary mechanisms [3,8]. So far, hundreds of Al-SR genes have been cloned in plants, represented by *AtSTOP1* in *Arabidopsis* and *OsART1* in rice [9,10,11,12,13,14,15,16,17]. However, as a crop sensitive to acidic soil [18], only a small number of Al-SR genes have been identified and functionally characterized in maize.

Here, we focus on the progress and perspective of Al-SR genes and their roles in the Al response in plants. Based on the cloned Al-SR genes, we propose the regulation mainly of networks of the Al response, utilizing *AtSTOP1* and *OsART1* as the key regulators in *Arabidopsis* and rice, respectively. Furthermore, we predict 166 putative Al-SR genes in maize based on orthologue and RNA-seq analyses. Moreover, we outline the potential strategies for alleviating Al stress in crop production, including crop rotation, the exogenous application of other elements, and molecular breeding.

## 2. Overview of Al-SR Genes in Plants

In *Arabidopsis* (76), rice (28), wheat (13), maize (8), and sorghum (5), at least 130 Al-SR genes have been cloned; however, compared to *Arabidopsis* and rice, fewer Al-SR genes have been functionally identified in maize (Figure 1A). To summarize the molecular mechanisms of the cloned Al-SR genes comprehensively, we classified these Al-SR genes into transporters, transcription factors, kinases/phosphatase, and those related to sugar metabolism, hormones, ROS metabolism, and other processes based on their functions, which contain 31, 30, 21, 8, 11, 10, and 19 genes, respectively (Figure 1B).

Among all the reported Al-SR genes, 90 were investigated for their protein subcellular localizations (Figure 1C). These proteins were localized in several organelles, such as the vacuole membrane/channel, vesicle membrane, plasma membrane, nucleus, etc. Among them, most proteins were localized in the nucleus (29), but fewer were localized in the Golgi (only one) (Figure 1C). These results indicate that the response to Al stress may take place in various organelles in plants.

Moreover, twelve Al-SR gene-encoding proteins were found to be localized in several organelles (e.g., the nucleus-, cytoplasm-, and endoplasmic reticulum-localized AtEIN2) (Figure 1C) [19]. SbSTAR1 [20,21], ZmMATE6 [22], OsMGT1 [23], OsASR1/5 [24], ZmALDH [25], AtNPR1 [19], SbGLU1 [20,21,26], and AtPP2C.D5/6/7 [27,28] were localized in the cytoplasm and nucleus, indicating that these genes may function in multiple organelles for Al stress.

Collectively, the protein subcellular localization information of Al-SR genes is largely consistent with their functions in the response to Al stress. Nevertheless, the detailed molecular mechanism of the response to Al stress is largely unclear and needs to be further investigated.

## 3. Al-SR Genes and Their Essential Roles in Plants

### 3.1. Transporters

Transporters are ubiquitous in all living organisms and constitute an integral component of the biological system [29]. In plants, there exists a diverse array of transporters, including ATP-binding cassette (ABC) transporters, multidrug and toxic compound extrusion (MATE) transporters, natural resistance-associated macrophage proteins (NRAMP), and so on [30].

Among the 31 Al-SR transporters, 8 ABC transporters have been identified (Table 1 and Appendix A). For example, AtSTAR1, the ortholog of OsSTAR1 and SbSTAR1, interacts with AtALS3. These are all involved in the basic detoxification of Al [20,21,31,32]. OsSTAR2 interacts with OsSTAR1, forming heterodimers in response to Al stress in rice [32]. AtALS1 and OsALS1 interact to sequestrate Al into the vacuoles [33,34]. ZmPGP1 is associated with reducing auxin accumulation in the root tips to regulate Al stress in maize [35]. Nine MATE transporters, such as AtMATE, increase Al resistance and improve carbon-use efficiency for Al resistance and AtFRDL3-mediated efflux of citrate into the root vasculature in *Arabidopsis* [36,37,38,39]. OsFRDL2 is involved in the Al-induced secretion of citrate, and OsFRDL4 responds to aluminum tolerance by enhancing its expression in rice [40,41,42]. SbMATE mediates Al-activated citrate efflux from the root apices in sorghum [20,43,44,45,46,47,48,49,50,51]. ZmMATE1 and ZmMATE2 are involved in citrate efflux in oocytes, as demonstrated in experiments on maize [52,53]. ZmMATE6 enhances Al tolerance in transgenic *Arabidopsis* [22]. TaMATE2 is related to Al tolerance in bread wheat [54]. ZmMATE1 is the ortholog of AtFRDL3, OsFRDL2, and TaMATE2, which play similar roles in Al-SR. Malate can regulate plant physiology, thereby facilitating the alleviation of Al-induced stress. There are five identified malate transporters, including AtALMT1 [36,37,55,56,57,58], AtALMT9 [59,60,61], AtALMT12 [62,63], OsALMT4 [63,64], and TaALMT1 [57,65]. These participate in malate transport in response to Al stress in plants. In addition, there are four metal transporters, including OsNrat1 [66,67,68], OsMGT1 [23], SbNrat1 [69], and ZmNRAMP4 [70]; two auxin transporters, including OsPIN2 [71,72] and OsAUX3 [73]; one oxalate transporter, called AtOT [74]; and two aquaporins, including AtNIP1;2 [75,76] and OsNIP1;2 [77]. These are closely correlated to the response to Al stress. Taken together, transporters play vital roles in material transport and are involved in Al-SR in plants.

### 3.2. Transcription Factor

The maize genome contains a total of 2216 protein-coding genes that have been predicted to be transcription factor (TF) genes [145]. Up to now, at least 30 Al-SR TF genes have been cloned in *Arabidopsis*, rice, sorghum, maize, and wheat (Table 1 and Appendix A), including 10 zinc finger TFs of AtSTOP1 [11,12,14,85] and AtSTOP2 [88] in *Arabidopsis*. OsART1 [9,10,15,16,17,34] and OsART2 [9] in rice, SbSTOP1a/b/c/d [86] and SbZNF1 [48] in sorghum, and TaSTOP1 [87] in wheat. Among them, AtSTOP1 and its orthologs in other plants, including OsART1, and SbSTOP1a/b/c/d, play common roles in Al stress by regulating other functional genes. The six WRKY TFs, including AtWRKY46, work as transcriptional repressors of AtALMT1 [89], and AtWRKY47 is involved Al stress via the regulation of cell wall-modifying genes [90] in *Arabidopsis*. OsWRKY22 promotes Al tolerance by the activation of OsFRDL4 in rice [42]. SbWRKY1, SbWRKY22, and SbWRKY65 positively regulate Al tolerance in sorghum [20,48]. The two abscisic acid, stress, ripening-induced (ASR) family TFs of OsASR1 and OsASR5 work as complementary transcription factors in regulating Al-responsive genes in rice [24,91,92]. The two HD-Zip TFs of AtHB7and AtHB12 respond to Al stress by regulating root growth in *Arabidopsis* [93], and one basic-leucine zipper (bZIP) TF of SbHY5 facilitates light-induced aluminum tolerance in sorghum by activating the expression of SbMATE and SbSTOP1s [146]. The two MYB TFs of AtMYB103 positively regulate Al sensitivity by mediating the modulation of the O-acetylation level of cell wall xyloglucan and act upstream of *TRICHOME BIREFRINGENCE-LIKE27* in *Arabidopsis* [96]. *OsMYB30* is regulated by OsART1 to response aluminum resistance in cell-wall modification in rice [95]. The two NAC TFs of ANAC017 regulate Al tolerance through the modulation of genes involved in cell-wall modification [97]. AtSOG1 suppresses growth reduction in plants under Al stress [98,99]. The JA signaling regulator of MYC2, a bHLH transcription factor, upregulates the response to Al stress of *Arabidopsis* root tips [100]. Additionally, another four TFs, including *AtLUH* [101,102], *AtSLK2* [101], *AtPIF4* [7], and *AtRBR1* [103], are also involved in Al tolerance in plants, indicating that these transcription factors may play core roles in plants under Al stress. However, further analysis is necessary for some TFs to gain a more comprehensive understanding, although the target genes of most TFs have been identified as responsive to Al stress.

### 3.3. Kinases/Phosphatase

Kinases and phosphatase play pivotal roles in plant stress response [146,147]. Up to now, at least 20 Al-SR kinases/phosphatase genes have been cloned in *Arabidopsis*, rice, sorghum, maize, wheat, and other plants (Table 1 and Appendix A). The cell wall-associated receptor kinase *AtWAK1* increases Al tolerance in terms of root growth [104]. The activity of AtCK2 kinase contributes to the development of Al toxicity tolerance, and regulates the DNA damage response (DDR) pathway by phosphorylating SOG1 [105]. The loss functions of *AtRAE1*, *AtRAE2*, *AtRAE3*/*AtHPR1*, and *AtRAH1* reduce Al resistance by acting as an E3 ligase to regulate the stability of the target proteins, such as AtSTOP1 and AtALMT1 [35,106,107,108]. However, the loss function of *AtESD4*/*RAE5* or *AtSIZ1* increases the transcriptional-level *AtALMT1*, thereby enhancing the resistance to Al in *atesd4/rae5* or *atsiz1* [109,111,148,149]. The AtMEKKK1-MKK1/2-MPK4 cascade plays a crucial role in Al signaling and confers resistance to Al by enhancing AtSTOP1 accumulation through phosphorylation-mediated mechanisms in *Arabidopsis* [112,150]. *OsSAL1*, a member of the PP2C.D family, is the ortholog of AtPP2C.D5/D6/D7 in *Arabidopsis*. Remarkably, both the *ossal1* mutant and the *atpp2c.d5/d6/d7* triple mutant exhibit more Al resistance compared to the WT, suggesting conserved yet complex roles of these phosphatases in modulating plant stress responses [27,28]. Additionally, OsSAL1 interacts with and dephosphorylates the plasma membrane H^+^-ATPase OsA7 to exert negative regulation on its function in Al stress [27]. AtATR phosphorylates AtSUV2 in vivo under Al stress [114]. In addition, the expression of certain genes is influenced by Al stress and other stress. For instance, the *atpah1/pah2* double mutant exhibits enhanced susceptibility to Al under low-phosphorus conditions [113]. The expression of *OsArPK*, an Al-related protein kinase gene, is induced in the roots following prolonged exposure to high concentrations of Al [115].

### 3.4. Sugar Metabolism

The cellular sugar status remains relatively stable under normal growth conditions but is adversely affected by various environmental perturbations [151,152]. In plants, at least eight Al-SR sugar metabolism-related genes have been cloned (Table 1 and Appendix A). *AtEXPA10* is an Al-inducible expansin gene that is regulated by AtART1 and plays an important role in modulating Al accumulation within root cell walls [116]. The expression of *ZmXTH* is significantly induced by Al toxicity, and the overexpression of *ZmXTH* in *Arabidopsis* enhances the tolerance to Al toxicity by reducing Al accumulation in both the roots and cell walls [117]. AtXTH15 and AtXTH31 are endo-trans-glucosylase-hydrolases and exhibit enhanced Al resistance in their mutants [118,119]. AtTBL27 influences the sensitivity of *Arabidopsis* to Al by modulating the Al-binding capacity in hemicellulose [96,120]. The identification of AtPME46 revealed its ability to reduce the binding of Al to cell walls, thereby alleviating Al-induced inhibition of root growth through the downregulation of PME enzyme activity [101]. Furthermore, the modified characteristics of hemicellulose contribute to its reduced Al accumulation in the *atparvus* mutant [121]. The β-1,3-glucanase SbGLU1 reduced callose deposition and increased tolerance to Al toxicity, highlighting the intricate interplay between cell wall components and aluminum stress responses in plants [20,26,122].

### 3.5. Hormone-Related Genes

Plant hormones occupy a central role in regulating essential aspects of growth, development, and adaptive responses to environmental stress [153]. At least 11 Al-SR hormone-related genes have been cloned in plants (Table 1 and Appendix A). For example, AtEIN2 and AtNPR1 are ethylene and salicylic acid signal factors. The loss functions of *AtEIN2* and *AtNPR1* display more susceptibility to Al stress than WT [19]. The local biosynthesis of auxin regulated by YUCs in the root apex transition zone mediates the inhibition of root growth in response to Al stress [7]. AtTAA1 is specifically upregulated in the root apex TZ in response to Al treatment [7,124]. Additionally, AtCOI1-mediated Al-induced root growth inhibition under Al stress was controlled by ethylene [100]. AtSUR1 and AtSUR2 promote IAA biosynthesis and auxin conjugation, respectively, and the *sur1* and *sur2* mutants exhibit increased sensitivity to Al stress [118,125,126].

### 3.6. ROS Metabolism

Reactive oxygen species (ROS) serve as crucial signaling molecules that facilitate prompt cellular responses to various stimuli in plants [154]. The production of ROS is significantly increased in plants under biotic or abiotic stresses, disrupting the homeostasis of -OH, O_2_^-^, and H_2_O_2_. To maintain the balance of ROS in vivo, some enzymes and low-molecular-weight compounds participate in antioxidant mechanisms in plants, including superoxide dismutases (SODs), catalases (CATs), ascorbate peroxidases (APx), glutathione peroxidases (GPx), ascorbic acid, glutathione, and tocoferol [155]. Up to now, at least 10 Al-SR hormone-related genes have been cloned in *Arabidopsis*, rice, sorghum, maize, and wheat (Table 1 and Appendix A).

In rice, H_2_O_2_ accumulation is significantly increased in *OsApx1/2*-silenced plants and presents higher Al tolerance than WT [127]. The overexpression of *AtGR* can maintain GSH levels, reinforcing the detoxification functions in plants and providing an efficient approach for enhancing Al tolerance [128]. The expressions of *AtGST1* and *AtGST11* are activated in response to Al stresses [129]. The *AtPrx64* gene increases root growth and mitigates the accumulation of Al and ROS in the roots [130]. *AtAOX1a* mitigates Al-induced programmed cell death (PCD) by preserving mitochondrial function and enhancing the expression of protective functional genes [131]. *ZmAT6* and *ZmALDH* confer Al tolerance via ROS scavenging and reduce Al accumulation in roots [25,132]. The involvement of *AtNADP-ME1* in regulating malate levels in the root apex leads to an elevation in the content of this organic acid [133]. In general, these ROS metabolism genes dynamically respond to aluminum stress by meticulously regulating ROS homeostasis, ensuring plant survival and resilience under adverse conditions.

### 3.7. Other Processes

Apart from the Al stress-related genes mentioned above, several additional genes have been reported to regulate Al stress response in plants (Table 1 and Appendix A). Examples include AtGRP3, which encodes a glycine-rich protein [134], *AtVHA-a2/a3*, which encodes a subunit of the vacuolar H^+^-ATPase (V-ATPase) [8], *AtSUV2*, a putative plant ATRIP homologue [114], and *AtALT1*, a thioesterase [78]. These negatively control Al stress in plants. *AtCBL1*, a calcineurin B-like calcium sensor [135], *AtALS7*, a ribosomal biogenesis factor [136], *AtSWA2*, a CCAAT-box binding factor [136], *AtRAD51*, a DNA repair family protein gene [103], *AtCYCB1*, a cyclin protein gene [103], *AtTANMEI*/*ALT2*, a WD40 protein gene [138], and *AtPGIP1*, a P450-dependent monooxygenase gene [139], positively regulate Al stress in *Arabidopsis*. OsGERLP [137], Os4CL3/4/5 [7,95,140,141,142], and OsCS1 [143] positively regulate Al stress in rice. Additionally, *TaWali1* and *TaWali5* positively regulate Al stress in wheat [144]. In a word, the response to Al stress is an intricate process, necessitating the coordination of multiple substances and genes.

## 4. The Primary Molecular Regulatory Network for the Cloned Al Stress-Related Genes in Plants

Plant response to Al stress is a fairly complicated process. Here, a molecular regulatory network for the cloned Al-SR genes in plants, which mainly include similar STOP1-related pathways in *Arabidopsis* and ART1-related pathways in rice, is summarized and updated, considering the functional properties (Figure 2).

### 4.1. STOP1-Related Pathway in Arabidopsis

*STOP1* (*SENSITIVE TO PROTEIN RHIZOTOXICITY 1*) is a zinc finger transcription factor that plays important roles in Al tolerance [11,12,14,54,85,86]. In *Arabidopsis*, *AtSTOP1* plays a central role in Al tolerance because of its ability to connect upstream kinases and downstream target genes (Figure 2). The AtMEKKK1-AtMKK1/2-AtMPK4 cascade exerts a positive regulatory effect on AtSTOP1 phosphorylation and stability. The phosphorylation of AtSTOP1 diminishes its interaction with the F-box protein AtRAE1 [112]. AtRAE1 interacts with and facilitates the ubiquitin-26S proteasome pathway-mediated degradation of AtSTOP1, while Al stress induces the accumulation of AtSTOP1 [6]. Meanwhile, AtRAH1, AtSIZ1, and AtESD4/RAE5 interact with AtSTOP1 and regulate AtSTOP1 SUMOylation under Al stress [106,109,149]. Additionally, AtRAE3 regulates AtSTOP1 mRNA exports under Al stress [107]. AtSTOP2 works as a physiologically minor isoform of AtSTOP1, and AtSTOP2 is directly regulated by AtSTOP1 [88]. In addition, AtSTOP1 regulates malate transporter gene *AtALMT1* [58], MATE transporter gene *AtMATE* [36,37], aquaporin gene *AtNIP1;2* [75,76], P450-dependent monooxygenase gene *AtPGIP1* [139], and ABC transporter gene *AtALS1*, and AtALS1 interacts with AtSTAR1 to form heterodimers [31].

### 4.2. ART1-Related Pathway in Rice

ART1 (Al resistance transcription factor 1), a C2H2-type zinc finger transcription factor, which is the ortholog of AtSTOP1, regulates the gene expressions associated with Al tolerance in rice [16]. OsART1 confers Al resistance by repressing the modification of cell wall properties regulated by OsMYB30, thereby enhancing the effect of Al resistance [95], and in turn repressing Os4CL5-dependent 4-coumaric acid accumulation, which is similar to the functions of Os4CL3 and Os4CL4 [7,140,141,142]. The MATE family protein genes of *OsFRDL2* and *OsFRDL4* are directly regulated by OsART1 and involved in the Al-induced secretion of citrate [40,41,42,80]. OsART1 directly regulates metal transporter gene *OsNRAT1*, and OsNRAT1 serves as the initial step in sequestering Al^3+^ into the vacuoles, thereby alleviating Al toxicity [66,67,68]. *OsEXPA10*, an Al-inducible expansion gene, is regulated by OsART1 and promotes Al accumulation in the root cell of rice [116]. Similar to AtSTOP1, OsART1 regulates OsSTAR1, which is orthologous with AtSTAR1. OsSTAR1 forms heterodimers with OsSTAR2 at tonoplasts [32]. In general, *AtSTOP1* and *OsART1* play pivotal roles in the response to Al stress in *Arabidopsis* and rice, making the STOP1/ART1-related pathways valuable models for studying Al stress in maize and other plant species.

## 5. Prediction of Putative Al Stress-Related Genes in Maize

Compared to *Arabidopsis* and rice, only eight maize Al stress-related genes have been identified in maize. Among them, five cloned Al stress-related genes encode transporters. For example, ZmPGP1, an ABCB transporter, mediated auxin efflux in an action, regulated Al stress, and was associated with reduced auxin accumulation in root tips [35,156]. ZmMATE1, ZmMATE2, and ZmMATE6 belong to the MATE family. Maize is Al-tolerant with a higher ZmMATE1 copy number; however, ZmMATE2 is involved in a novel Al-tolerance mechanism [52,53,79]. ZmMATE6 displays a greater Al-activated release of citrate from the roots and is significantly resistant to Al toxicity [22]. ZmNRAMP4 is a metal transporter that enhances Al tolerance via the cytoplasmic sequestration of Al in maize [70]. Translocating the expression of ZmXTH, a xyloglucan endotransglucosylase/hydrolase gene, enhances tolerance to Al toxicity by reducing the Al accumulation in the roots and cell wall in *Arabidopsis* [117]. Two Al stress-related genes belong to ROS metabolism genes. For example, ZmAT6 confers Al tolerance via ROS scavenging [132]. ZmALDH participates in Al-induced oxidative stress and Al accumulation in roots [25]. To discover more Al stress-related genes in maize, putative Al stress-related genes in maize are predicted based on ortholog analysis and maize root RNA-seq analyses. Here, a total of 166 putative maize genes associated with Al stress were identified by analyzing the orthologs of other plants based on the Ensembl Plants website (https://plants.ensembl.org/index.html, accessed on 26 February 2024). Those 166 putative Al stress-related genes in maize are distributed among the ten chromosomes of maize with variable numbers, from twelve on chromosome 6 and chromosome 10 to twenty-eight on chromosome 2 (Figure 3, Appendix A). The in silico mapping information can facilitate gene cloning and evolutionary studies of the Al stress-related genes in maize.

## 6. Potential Applications to Alleviate Al Stress in Crop Production

The toxicity of Al poses a global challenge in acidic soils (pH < 5.5), leading to diminished crop growth and reduced productivity [1]. Previous studies have shown that Al have pleiotropic functions of beneficial or toxic effect to plants and other organisms, depending on factors such as the metal concentration, the chemical form of Al, the growth conditions, and the plant species [157]. Consequently, alleviating Al stress and even harnessing Al resources efficiently is imperative for sustainable agricultural production. To mitigate Al stress, we propose potential applications to alleviate Al stress in crop production based on the current research (Figure 4).

In previous studies, crop rotation has been considered as an effective way to alleviate heavy-metal stress [158]. Implementing a crop rotation strategy that involves the selection of low Al-accumulating cultivars, along with effective water and manure management practices, to achieve the purpose of soil improvement, can potentially serve as an efficacious approach to mitigate Al-induced damage (Figure 4). Additionally, applying other exogenous elements in crop growth is also a viable method (Figure 4). For example, the alleviation of Al toxicity by H_2_S is associated with an increase in ATPase activity, as well as a reduction in Al uptake and oxidative stress in barley at the seedling stage [159]. The uptake of NH^4+^ leads to a decrease in pH, which in turn alters the properties of the cell wall and reduces the Al accumulation by NH^4+^-induced mechanisms, rather than through direct competition for binding sites between Al^3+^ and NH^4+^ [84]. The application of exogenous Si treatment results in the formation of hydroxy Al silicates within the apoplast of the root apex, thereby effectively detoxifying Al [160]. For breeders, the issue of crop Al toxic needs to be solved from the original source, such as the development of new Al-tolerant varieties by using molecular breeding techniques (Figure 4). In summary, it is imperative to explore more efficient and convenient approaches in order to alleviate the detrimental effects of Al stress on crop production, aiming for enhanced quality and yield.

## 7. Conclusions and Perspectives

Al stress is a significant hazard in plant growth in low-pH environments, and thus, it affects organ development and ultimately reduces the grain yield in crops [161]. Here, we systematically investigated the Al-SR genes and their roles in controlling the response of plants to Al. To date, most of the cloned Al-SR genes have been identified in *Arabidopsis* and rice, with a number of genes reported in maize (only eight). Here, we predicted 166 maize orthologs of Al-SR genes in other plants and determined their precise chromosome localizations in the maize genome (Figure 3). This research provides a batch of targets genes to study the molecular mechanisms and genetic improvement of the Al response of maize by using CRISPR/Cas9 mutagenesis or other biotechnologies. In acidic soil conditions, even trace amounts of Al can elicit severe and irreversible toxicity symptoms in higher plants, drastically hindering water and nutrient uptake, and thereby imposing considerable stress on plant growth [4]. Therefore, we provide some potentially effective applications for mitigating Al stress in crop production, aiming to cultivate healthy and high-yielding crops even under the challenging conditions imposed by Al toxicity. Therefore, the investigation of the functional mechanisms of Al-SR genes and the exploration of new methods to mitigate Al stress are formidable tasks to enhance the crop grain yield. These tasks should be given priority considerations in future work.

## Figures and Tables

**Figure 1 ijms-25-09045-f001:**
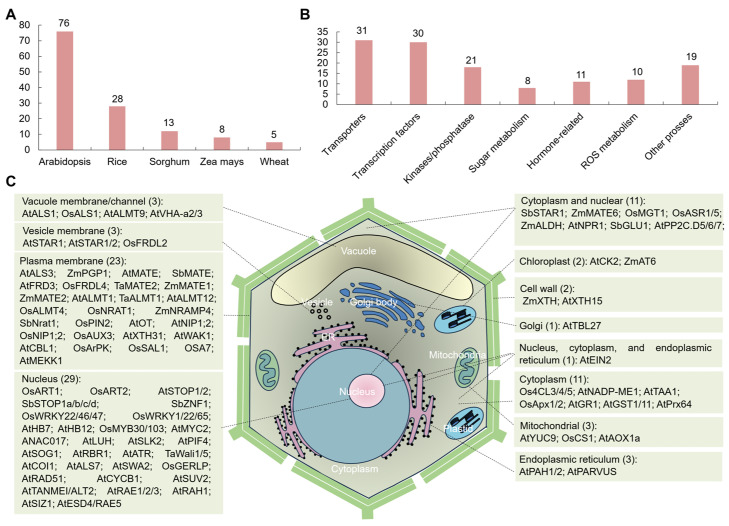
Identified aluminum stress-related genes, their subcellular localizations, and their roles in plants, and the expression analysis of the cloned maize aluminum stress-related genes in different developmental stages of maize roots. (**A**) The cloned aluminum stress-related genes in *Arabidopsis*, rice, maize, wheat, and sorghum. (**B**) Classification of the cloned aluminum stress-related genes into transporters, transcription factors, kinases/phosphatase, and those related to sugar metabolism, hormones, ROS metabolism, and other processes. (**C**) The protein subcellular localizations of the aluminum stress-related genes in plants.

**Figure 2 ijms-25-09045-f002:**
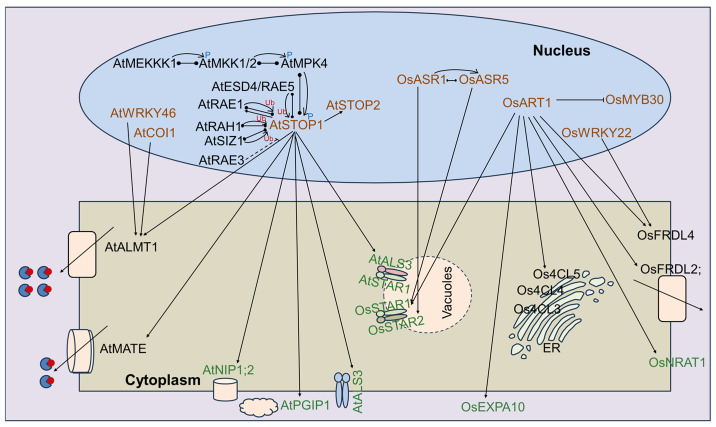
The primary signaling pathways of the cloned aluminum stress-related genes involved in plants.

**Figure 3 ijms-25-09045-f003:**
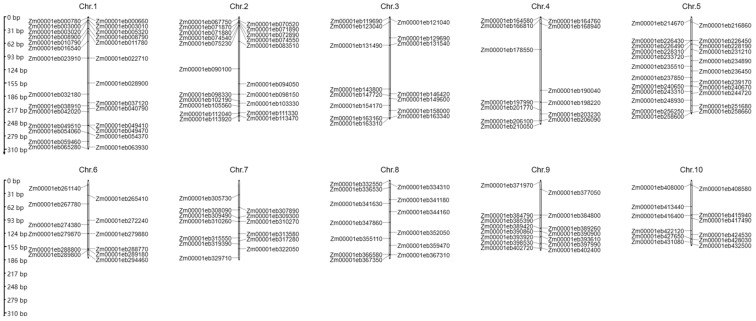
The precise chromosomal locations of the 166 predicted aluminum stress-related genes in the maize genome.

**Figure 4 ijms-25-09045-f004:**
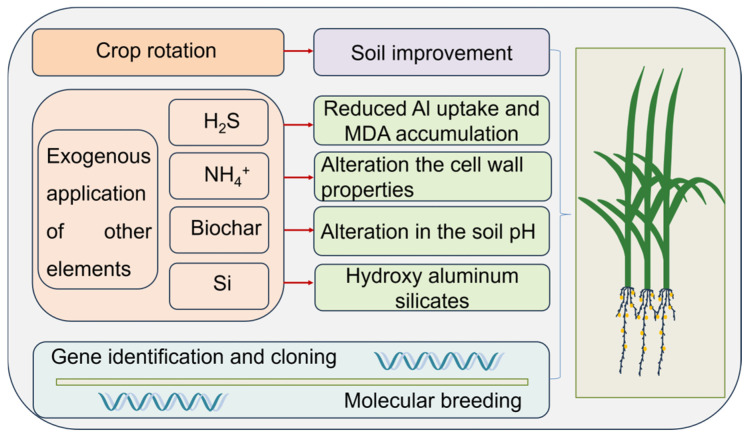
Potential applications to alleviate aluminum stress in crop production.

**Table 1 ijms-25-09045-t001:** Functional classifications of the reported Al-SR genes in *Arabidopsis*, rice, wheat, maize, and sorghum.

No.	Gene Names	Gene ID	Encoded Proteins	Biological Function	References
**Transporters**					
1	*AtSTAR1*	At1g67940	ABC transporter	AtSTAR1 is involved in the basic detoxification of Al in *Arabidopsis*.	[31]
2	*OsSTAR1*	Os06g0695800	ABC transporter	OsSTAR1 interacts with OsSTAR2 and is involved in the detoxification of Al.	[32]
3	*SbSTAR1*	SORBI_3010G246200	ABC transporter	*SbSTAR1* enhances Al tolerance by regulating hemicellulose content in the root cell wall.	[20,21]
4	*OsSTAR2*	Os05g0119000	ABC transporter	OsSTAR1 interacts with OsSTAR2 and is involved in the detoxification of Al.	[32]
6	*AtALS1*	At5g39040	ABC transporter	Contributes to Al redistribution between the cytoplasm and vacuoles and to symplastic Al detoxification.	[33]
7	*OsALS1*	Os03g0755100	ABC transporter	Responsible for sequestrating Al into vacuoles.	[34]
5	*AtALS3*	At2g37330	ABC transporter	Required for Al resistance/tolerance and distribution to gather Al away from sensitive tissues to protect the growing root from the toxic effects of Al.	[78]
8	*ZmPGP1*	Zm00001eb038710	ABC transporter	*ZmPGP1* regulates Al stress and is associated with reduced auxin accumulation in root tips.	[35]
9	*AtMATE*	At5g52450	Multidrug and toxic compound extrusion (MATE) family protein	Increased Al resistance of the transgenic plants and enhanced carbon-use efficiency for Al resistance.	[36,37]
10	*SbMATE*	SORBI_3009G106960	Multidrug and toxic compound extrusion (MATE) family protein	*SbMATE* is associated with the induction of Al tolerance via enhanced root citrate exudation.	[20,43,44,45,46,47,49,50,51]
11	*ZmMATE1*	Zm00001eb261140	Multidrug and toxic compound extrusion (MATE) family protein	Maize lines with a higher *ZmMATE1* copy number are Al-tolerant.	[52,53,79]
12	*AtFRDL3*	At3g08040	Multidrug and toxic compound extrusion (MATE) family protein	*AtFRD3* confers tolerance to aluminum.	[38,39]
13	*OsFRDL2*	Os10g0206800	Multidrug and toxic compound extrusion (MATE) family protein	*OsFRDL2* is involved in the Al-induced secretion of citrate.	[40]
14	*TaMATE2*	TraesCS1A02G305200, TraesCS1B02G315900, TraesCS1D02G304800	Multidrug and toxic compound extrusion (MATE) family protein	*TaMATE2* is involved in Al tolerance in bread wheat.	[54]
15	*ZmMATE2*	Zm00001eb219790	Multidrug and toxic compound extrusion (MATE) family protein	*ZmMATE2* is involved in a novel Al tolerance mechanism.	[79]
16	*ZmMATE6*	Zm00001eb230490	Multidrug and toxic compound extrusion (MATE) family protein	*ZmMATE6* displays a greater Al-activated release of citrate from the roots and is significantly resistant to Al toxicity.	[22]
17	*OsFRDL4*	Os01g0919100	Multidrug and toxic compound extrusion (MATE) family protein	*OsFRDL4* protein was able to transport citrate and was activated by Al.	[41,42,80]
18	*AtALMT1*	At1g08430	Malate transporter	*AtALMT1* confers acid–soil tolerance by releasing malate from roots and enhances the response to trivalent cations.	[36,37,55,56,57,58]
19	*TaALMT1*	TraesCS2A02G297900	Malate transporter	*TaALMT1* confers acid–soil tolerance by releasing malate from roots, enhances response to trivalent cations, and is permeable not only to malate but also to other physiologically relevant anions.	[57,62,65,81,82,83]
20	OsALMT4	Os01g0221600	Malate transporter	*OsALMT4* facilitates malate efflux from cells and protects plants from Al stress, and its expression is altered in low-light environments.	[63,64]
21	*AtALMT9*	At3g18440	Malate transporter	AtALMT9 is a tetramer, and the TMa5 domains of its subunits contribute to form the pores of anion channels.	[60,61]
22	*AtALMT12*	At4g17970	Malate transporter	An anion transporter involved in stomatal closure.	[62,63]
23	*OsNrat1*	Os02g0131800	Metal transporter	The preliminary step to sequester Al^3+^ into vacuoles and thus relieve Al toxicity.	[66,67,68]
24	*SbNrat1*	SORBI_3004G029900	Metal transporter	Selectively transports Al^3+^ and is involved in basic Al tolerance in sorghum.	[69]
25	*ZmNRAMP4*	Zm00001d015133	Metal transporter	ZmNRAMP4 enhances Al tolerance via cytoplasmic sequestration of Al in maize.	[70]
26	*OsMGT1*	Os01g0869200	Magnesium transporter	Al induces the upregulation of *OsMGT1* to increase the Mg content in cells, thereby preventing the binding of Al to enzymes and other cellular components and enhancing the aluminum tolerance of rice.	[23]
27	*OsPIN2*	Os06g0660200	Auxin transporter	Overexpression of *OsPIN2* altered the distribution of Al^3+^ in apical cells, as indicated by a significant increase in the content of Al^3+^ in the cytosol and a decrease in the cell wall.	[71,72]
31	*OsAUX3*	Os05g0447200	Auxin transporter	Involved in Al-induced inhibition of root growth.	[73]
28	*AtOT*	At4g09580	Oxalate transporter	Oxalate involves AtOT to enhance oxalic acid resistance and aluminum tolerance.	[74]
29	*AtNIP1;2*	At4g18910	Aquaporins	AtNIP1;2 mediates Al uptake and demonstrates critical roles of the constriction regions for transport activities.	[75,84]
30	*OsNIP1;2*	Os01g0202800	Aquaporins	OsNIP1;2 confers internal Al detoxification via taking out the root cell wall’s Al, sequestering it to the root cell’s vacuole, and re-distributing it to the above-ground tissues.	[77]
**Transcription factors**					
1	*OsART1*	Os12g0170400	Zinc finger transcription factor	Regulates the expression of genes related to Al tolerance in rice.	[9,10,15,16,17]
2	*OsART2*	Os04g0165200	Zinc finger transcription factor	The expression of *OsART2* is rapidly induced by Al in the roots of wild-type rice, and the knockout of OsART2 increases sensitivity to Al toxicity.	[9]
3	*AtSTOP1*	At1g34370	Zinc finger transcription factor	*AtSTOP1* binding to the consensus motif in the promoters of *AtSTOP2*, *AtALMT1*, *AtGDH1*, and *AtGDH2* with high affinity to drive their expression. Fe^2/3+^ and Al^3+^ act similarly to increase the stability of STOP1 and its accumulation in the nucleus, where it activates the expression of *AtALMT1*.	[11,12,14,85]
4	*SbSTOP1a*	SORBI_3001G020200	Zinc finger transcription factor	SbSTOP1 plays an important role in Al tolerance in sweet sorghum and extends our understanding of the complex regulatory mechanisms of STOP1-like proteins in response to Al toxicity.	[86]
5	*SbSTOP1b*	SORBI_3004G188300	Zinc finger transcription factor		
6	*SbSTOP1c*	SORBI_3007G166000	Zinc finger transcription factor		
7	*SbSTOP1d*	SORBI_3003G370700	Zinc finger transcription factor		
8	*TaSTOP1*	TraesCS3A02G381900, TraesCS3B02G414500, TraesCS3D02G375000	Zinc finger transcription factor	TaSTOP1 could be a potential candidate gene for genomic-assisted breeding for Al tolerance in bread wheat.	[87]
9	*AtSTOP2*	At5g22890	Zinc finger transcription factor	STOP2 is a physiologically minor isoform of STOP1 and activates the expression of genes regulated by STOP1.	[88]
10	*SbZNF1*	SORBI_3009G151400	Zinc finger transcription factor	SbWRKY1 and SbZNF1 transcriptional activation of *SbMATE*.	[48]
11	*SbWRKY1*	SORBI_3009G174300	WRKY transcription factor		
12	*OsWRKY22*	Os01g0820400	WRKY transcription factor	OsWRKY22 promotes Al-induced increases in *OsFRDL4* expression, thus enhancing Al-induced citrate secretion and Al tolerance in rice.	[42]
13	*AtWRKY46*	At2g46400	WRKY transcription factors	Regulating aluminum-induced malate secretion.	[89]
14	*AtWRKY47*	At4g01720	WRKY transcription factor	*WRKY47* is required for root growth under both normal and Al stress conditions via direct regulation of cell wall modification genes.	[90]
15	*SbWRKY22*	SORBI_3002G418500	WRKY transcription factor	OE-*SbWRKY22/65* plants enhance Al tolerance by reducing callose deposition in roots.	[20]
16	*SbWRKY65*	SORBI_3003G285500	WRKY transcription factor		
17	*OsASR1*	Os02g0543000	ASR (abscisic acid, stress, ripening-induced) family transcription factors	ASR1 and ASR5 act in concert and complementarily regulate gene expression in Al response.	[24]
18	*OsASR5*	Os11g0167800	ASR (abscisic acid, stress, ripening-induced) family transcription factors	OsASR5 is sequestered in the chloroplasts as an inactive transcription factor that could be released to the nucleus in response to Al to regulate genes related to photosynthesis.	[24,91,92]
19	*AtHB7*	At2g46680	HD-Zip I transcription factors	AtHB7 and AtHB12 oppositely regulate Al resistance by enacting Al accumulation in root cell walls, enabling homodimers or heterodimers in response to Al stress.	[93]
20	*AtHB12*	At3g61890	HD-Zip I transcription factors		
21	*SbHY5*	SORBI_3004G085600	Basic-leucine zipper (bZIP) transcription factor family protein	SbHY5 confers Al tolerance in plants by modulating Al-SR gene expression.	[94]
22	*OsMYB30*	Os09g0431300	MYB transcription factor	OsART1 confers Put-promoted Al resistance via the repression of OsMYB30-regulated modification of cell wall properties in rice.	[95]
23	*AtMYB103*	At1g63910	MYB transcription factor	AtMYB103 acts upstream of AtTBL27 to positively regulate Al resistance by modulating the O-acetylation of the cell wall XyG.	[96]
24	*AtNAC017*	At1g34190	NAC transcription factors	Regulates Al tolerance in *Arabidopsis* by positively regulating the expression of *AtXTH31*.	[97]
25	*AtSOG1*	At1g25580	NAC transcription factors	Suppressed growth reduction in plants on Al-containing media. *sog1* mutants are sensitive to Al.	[98,99]
26	*AtMYC2*	At1g32640	bHLH transcription factor	Upregulated in response to Al stress in root tips.	[100]
27	*AtLUH*	At2g32700	Groucho-like family of transcriptional corepressor	Promotes Al accumulation in the root cell wall.	[101,102]
28	*AtSLK2*	At5g62090	SEUSS-like	The *atslk2* mutants responded to Al in a similar way as LUH mutants, suggesting that a LUH–SLK2 complex represses the expression of *AtPME46*.	[101]
29	*AtPIF4*	At2g43010	Phytochrome interacting factor	AtPIF4 promotes Al-inhibited primary root growth by regulating the local expression of YUCs and auxin signal in the root apex TZ.	[7]
30	*AtRBR1*	At3g12280	Retinoblastoma protein	RBR1 is targeted to DNA break sites in a CDKB1 activity-dependent manner and partially co-localizes with RAD51 at damage sites.	[103]
**Kinases/phosphatase**					
1	*AtWAK1*	At1g21250	Cell wall-associated receptor kinase	OE-*AtWAK1* shows an enhanced Al tolerance in terms of root growth.	[104]
2	*AtCK2*	At4g17640	Casein kinase	AtCK2 controls the DDR pathway through phosphorylation of SOG1.	[105]
3	*AtRAE1*	At5g01720	F-box protein	RAH1 and/or RAE1 participate in the regulation of Al resistance and plant growth, and also function as an E3 ligase in the regulation of STOP1 stability.	[6,106]
6	*AtRAH1*	At5g27920	F-box protein		[106]
4	*AtRAE2*	At5g56130	Core component of the THO complex	The *atrae2* mutant is less sensitive to Al; RAE2 regulates AtALMT1 and modulates low Pi response.	[107]
5	*AtRAE3/AtHPR1*	At5g09860	THO/TREX complex	Mutation of RAE3 reduces Al resistance and low phosphate response.	[107,108]
7	*AtESD4/RAE5*	At4g15880	SMALL UBIQUITIN-LIKE MODIFIER	Mutation of *ESD4* increases the level of STOP1 SUMOylation.	[109]
8	*AtSIZ1*	At5g60410	SUMO E3 ligase	AtSIZ1 regulates Al resistance and low Pi response through the modulation of AtALMT1 expression. “SIZ1–STOP1–ALMT1” is involved in root growth response to Al stress.	[106,109,110,111]
9	*AtMEKK1*	At4g08500	Mitogen-activated protein kinase (MAPK) kinase kinase kinases	MEKK1-MKK1/2-MPK4 cascade is important for Al signaling and confers Al resistance through phosphorylation-mediated enhancement of STOP1 accumulation in *Arabidopsis*.	[112]
10	*AtMKK1*	At4g26070	MAP kinase kinases		
11	*AtMKK2*	At4g29810	MAP kinase kinases		
12	*AtMPK4*	At4g01370	MAP kinases		
13	*OsSAL1*	Os06g0717800	PP2C.D phosphatase	*osals1* increased PM H^+^-ATPase activity and Al uptake, causing hypersensitivity to internal Al toxicity.	[27]
14	*AtPP2C.D5*	At4g38520	PP2C.D phosphatase	The *atpp2c.d5d6d7* triple mutant was more resistant to Al than WT.	[27,28]
15	*AtPP2C.D6*	At3g51370	PP2C.D phosphatase		
16	*AtPP2C.D7*	At5g66080	PP2C.D phosphatase		
17	*OsA7*	Os04g0656100	H+-ATPase	OsSAL1 interacts with OsA7 to negatively regulate the PM H^+^-ATPase function.	[27]
18	*AtPAH1*	At3g09560	Phosphatidate phosphatase	The *pah1/pah2* double mutant shows enhanced Al susceptibility under low-P conditions.	[113]
19	*AtPAH2*	At5g42870	Phosphatidate phosphatase		
20	*AtATR*	At5g40820	Plant ATRIP	SUV2 may be a phosphorylation target of ATR.	[114]
21	*OsArPK*	Os06g0693000	Al-related protein kinase	*OsArPK* expression is induced by longer exposure to a high Al concentration in the roots.	[115]
**Sugar metabolism**					
1	*OsEXPA10*	Os04g0583500	Al-inducible expansin gene	The root cell wall of the knockout lines accumulated less Al than that in the wild type.	[116]
2	*ZmXTH*	Zm00001eb414340	Xyloglucan endotransglucosylase/hydrolase	Overexpression of ZmXTH in *Arabidopsis* enhanced its tolerance to Al toxicity by reducing Al accumulation in its roots and cell wall.	[117]
3	*AtXTH15*	At4g14130	Xyloglucan endotransglucosylase/hydrolases	The *atxth15* showed enhanced Al resistance.	[118]
4	*AtXTH31*	At3g44990	Endotransglucosylase-hydrolase	AtXTH31 affects Al sensitivity by modulating cell wall xyloglucan content and Al binding capacity.	[119]
5	*AtTBL27*	At1g70230	XyG O-acetyltransferase	Modulation of the O-acetylation level of XyG influences the Al sensitivity in *Arabidopsis* by affecting the Al-binding capacity in hemicellulose.	[96,120]
6	*AtPME46*	At5g04960	Pectin methylesterase	AtPME46 was found to reduce Al binding to cell walls and alleviate Al-induced root growth inhibition by decreasing PME enzyme activity.	[101]
7	*AtPARVUS*	At1g19300	Glucuronoxylan	The altered properties of hemicellulose contribute to decrease Al accumulation in *parvus* mutant.	[121]
8	*SbGLU1*	SORBI_3002G402700	β-1,3-glucanase enzyme	β-1,3-glucanase reduced callose deposition and increased tolerance to aluminum toxicity.	[20,26,122]
**Hormone-related**					
1	*AtEIN2*	At5g03280	Ethylene signaling	Double mutant *ein2-1/npr1-1* displayed more sensitivity to Al stress than wild-type plants.	[19]
2	*AtYUC9*	At1g04180	Flavin monooxygenase-like protein	YUCs regulated local auxin biosynthesis in the root apex TZ, mediating root growth inhibition in response to Al stress.	[123]
3	*AtYUC8*	At4g28720	Flavin monooxygenase-like protein		
4	*AtYUC7*	At2g33230	Flavin monooxygenase-like protein		
5	*AtYUC3*	At1g04610	Flavin monooxygenase-like protein		
6	*AtYUC5*	At5g43890	Flavin monooxygenase-like protein		
7	*AtTAA1*	At1g70560	Trp aminotransferase	TAA1 is specifically upregulated in the root apex TZ in response to Al treatment.	[7,124]
8	*AtCOI1*	At2g39940	Coronatine-insensitive	AtCOI1-mediated Al-induced root growth inhibition under Al stress controlled by ethylene.	[100]
9	*AtSUR1*	At2g20610	Tyrosine transaminase family protein	SUR1 promotes IAA biosynthesis via the indole-3-acetaldoxime pathway, *superroot2*, and *superroot1* mutant increased Al sensitivity.	[118,125]
10	*AtSUR2*	At4g31500	Cytochrome P450 CYP83B1	SUR2 may be involved in the control of auxin conjugation, and the *superroot2* and superroot1 mutant had increased aluminum sensitivity.	[118,126]
11	*AtNPR1*	At1g64280	Regulatory protein	Double mutant *ein2-1/npr1-1* displayed more sensitivity to Al stress than wild-type plants.	[19]
**ROS metabolism**					
1	*OsApx1*	Os03g0285700	Ascorbate peroxidases	Apx1/2-silenced plants also showed increased H_2_O_2_ accumulation under control and stress situations and presented higher tolerance to a toxic concentration of Al when compared to WT.	[127]
2	*OsApx2*	Os07g0694700	Ascorbate peroxidases		
3	*AtGR1*	At3g24170	Glutathione reductase	*GR*, an efficient approach to enhance Al tolerance, maintained GSH and reinforced dual detoxification functions in plants.	[128]
4	*AtGST1*	At1g02930	Glutathione S-transferase	Gene expression in response to Al stresses.	[129]
5	*AtGST11*	At1g02920	Glutathione S-transferase		
6	*AtPrx64*	At5g42180	Peroxidases	The *AtPrx64* gene increases the root growth and reduces Al accumulation and ROS in roots.	[130]
7	*AtAOX1a*	At3g22370	Alternative oxidase	*AtAOX1a* alleviates Al-induced PCD by maintaining mitochondrial function and promoting the expression of protective functional genes.	[131]
8	*ZmAT6*	Zm00001eb154120	-	*ZmAT6* confers aluminum tolerance via reactive oxygen species scavenging.	[132]
9	*ZmALDH*	Zm00001d017418	Aldehyde dehydrogenase	ZmALDH participates in Al-induced oxidative stress and Al accumulation in roots.	[25]
10	*AtNADP-ME1*	At2g19900	NADP-dependent malic enzyme	NADP-ME1 is involved in adjusting the malate levels in the root apex, and its loss results in an increased content of this organic acid.	[133]
**Other processes**					
1	*AtGRP3*	At2g05520	Glycine-rich protein	AtGRP3 functions in root size determination during development and in Al stress.	[134]
2	*AtCBL1*	At4g17615	Calcineurin B-like calcium sensors	Mutation of CBL1 suppresses root malate efflux.	[135]
3	*AtALS7*	At1g72480	Ribosomal biogenesis factor	The *atals7–1* is related to the expression of the S-adenosylmethionine recycling factor and reduced levels of endogenous polyamines.	[136]
4	*AtSWA2*	At1g72440	CCAAT-box binding factor	*AtSWA2* is required for normal gametogenesis and mitotic progression.	[136]
5	*OsGERLP*	Os03g0168900	Ribosomal L32-like protein	Low expression of *OsGERLP* caused the gene-silenced rice to be sensitive to Al, while high expression induced the Al tolerance in transgenic tobacco.	[137]
6	*AtVHA-a2*	At2g21410	Subunit of the vacuolar H+-ATPase (V-ATPase)	The *vha-a2 vha-a3* mutants displayed less sensitivity with lower Al accumulation in the roots compared to the wild-type plants when grown under excessive Al^3+^.	[8]
7	*AtVHA-a3*	At4g39080	Subunit of the vacuolar H+-ATPase (V-ATPase)		
8	*AtRAD51*	At3g22880	DNA repair (Rad51) family protein	RBR1 targets DNA break sites in CDKB1-CYCB1 complexes in an activity-dependent manner and partially co-localizes with RAD51 at damage sites.	[103]
9	*AtCYCB1*	At4g37490	Cyclin		
10	*AtSUV2*	At5g45610	Putative plant ATRIP homolog	Loss of *SUV2* reverses hypersensitivity of *als3-1* to Al. SUV2 detects Al damage in an ATR-dependent manner and is required for Al-dependent cell cycle arrest and terminal differentiation.	[114]
11	*AtTANMEI/ALT2*	At4g29860	WD40 protein	ALT2 is required for active stoppage of root growth after Al exposure.	[138]
12	*AtPGIP1*	At5g06860	P450-dependent monooxygenases	Involved in STOP1-dependent regulation in phosphoinositide signaling pathway, and regulates *PGIP1* expression under Al stress	[139]
13	*AtALT1*	At1g35290	Thioesterase	The *alt1* mutant positively impacts Al resistance in a manner dependent on pH adjustment.	[78]
14	*OsRAL1/4CL4*	Os06g0656500	4-Coumarate: coenzyme A ligase	4-coumaric acid and ferulic acid reduce Al binding to hemicellulose and consequently enhances Al resistance in *ral1/4cl4* mutants.	[140,141]
15	*Os4CL3*	Os02g0177600	4-Coumarate: coenzyme A ligase	4CL3 is involved in the regulation of lignin accumulation and Al resistance.	[140,142]
16	*Os4CL5*	Os08g0448000	4-Coumarate: coenzyme A ligase	Enhances resistance of os4cl5 mutant to Al.	[142]
17	*OsCS1*	Os02g0194100	Citrate synthase	OsCS1 is induced by Al toxicity	[143]
18	*TaWali1*	TraesCS1A02G115900	-	The Tawali1 and Tawali5 mutants have a generalized response for Al stress.	[144]
19	*TaWali5*	TraesCS1D02G265800	-		

## Data Availability

All data are shown in the main manuscript and in the Appendix A.

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
