# Peer review of "Systematic Investigation of Aluminum Stress-Related Genes and Their Critical Roles in Plants"

_ijms, 2024, doi:10.3390/ijms25169045_

Round 1

Reviewer 1 Report

Comments and Suggestions for Authors

Dear Authors,

Reviewer comments ijms-3153797

The review manuscript entitled „Systematic investigation of aluminum stress-related genes and their critical roles in plants“ represents a valuable overview focused on aluminum stress-related genes in plants including transporters, transcription factors, signaling proteins including kinases and phosphatases, sugar metabolism-related proteins, ROS-related, and other processes related proteins. The major plant species where these proteins are discussed include Arabidopsis, rice, and maize. I can recommend the review manuscript for publication in IJMS due to the importance of the topic; however, I have some important comments on the present manuscript:

1/ Figure 2: I think that Figure 2 providing just a schematic overview on cloned aluminum stress-related genes in plants and contains just information provided in Table 1 in a greater detail. Thus I think that Figure 2 is not necessary and should be removed.

2/ Terminology:  The spelling of some important terms has to be correcetd, e.g., „ROS-related“ and „other processes“ instead of „Ros-relatedů and „other prosses“ both in Table 1 and the manuscript text.

3/ Other formal comments on the text:

Line 55: Correct the typing error in the term „other processes“ (not „other prosses“).

Line 89: Modify the statement as follows: „These results indicate that response to Al stress in plants may be underlied by participation of multiple organelles.“

Line 159: Add „a“ prior to the words „common role“, i.e., „…play a common role in Al stress…“

Line 161: Correct the typing error in the term „Al stress“ (not „al stress“).

Line 179: Add „a“ prior to the words „core role“ in the statement: „Indicating that those transcription factors may play a core role in plants under Al stress.“

Line 202: Use the words „enhanced susceptibility“ instead of „heightened susceptibility“.

Line 210: Add „an“ preceding the words „important role“ i.e., „…play an important role in Al accumulated in root cell wall…“

Line 213: Correct the typing error in the chemical name „endo- trans-glucosylase-hydrolase“ (not „edotransglucosylase-hydrolase“).

Line 234: Write „ROS-related proteins“ with capital letters in „ROS“ (not „Ros“).

Line 256: Correct the typing error in the heading „3.7. Other processes“ (not „Other prosses“).

Line 402: Renumber the part „Conclusions and perspectives“ as part 7. instead of 6. since part 6.“ The potential applications of alleviating Al stress in crop production“ is given on lines 368-401, i.e., prior to part 7. Conclusions and perspectives.

Conclusions and perspectives, line 417: Modify the word form „explore“ to „exploration“ and remove the words „practical application values“ in the statement: „Therefore, investigation of functional mechanism of Al-SR genes, and exploration new methods to mitigating Al stress are formidable tasks to enhance crop grain yield.“ „These taska should be given priority considerations in future work.“

Final recommendation: Accept after a minor revision.

Comments on the Quality of English Language

Dear Authors,

Reviewer comments ijms-3153797

The review manuscript entitled „Systematic investigation of aluminum stress-related genes and their critical roles in plants“ represents a valuable overview focused on aluminum stress-related genes in plants including transporters, transcription factors, signaling proteins including kinases and phosphatases, sugar metabolism-related proteins, ROS-related, and other processes related proteins. The major plant species where these proteins are discussed include Arabidopsis, rice, and maize. I can recommend the review manuscript for publication in IJMS due to the importance of the topic; however, I have some important comments on the present manuscript:

1/ Figure 2: I think that Figure 2 providing just a schematic overview on cloned aluminum stress-related genes in plants and contains just information provided in Table 1 in a greater detail. Thus I think that Figure 2 is not necessary and should be removed.

2/ Terminology:  The spelling of some important terms has to be correcetd, e.g., „ROS-related“ and „other processes“ instead of „Ros-relatedů and „other prosses“ both in Table 1 and the manuscript text.

3/ Other formal comments on the text:

Line 55: Correct the typing error in the term „other processes“ (not „other prosses“).

Line 89: Modify the statement as follows: „These results indicate that response to Al stress in plants may be underlied by participation of multiple organelles.“

Line 159: Add „a“ prior to the words „common role“, i.e., „…play a common role in Al stress…“

Line 161: Correct the typing error in the term „Al stress“ (not „al stress“).

Line 179: Add „a“ prior to the words „core role“ in the statement: „Indicating that those transcription factors may play a core role in plants under Al stress.“

Line 202: Use the words „enhanced susceptibility“ instead of „heightened susceptibility“.

Line 210: Add „an“ preceding the words „important role“ i.e., „…play an important role in Al accumulated in root cell wall…“

Line 213: Correct the typing error in the chemical name „endo- trans-glucosylase-hydrolase“ (not „edotransglucosylase-hydrolase“).

Line 234: Write „ROS-related proteins“ with capital letters in „ROS“ (not „Ros“).

Line 256: Correct the typing error in the heading „3.7. Other processes“ (not „Other prosses“).

Line 402: Renumber the part „Conclusions and perspectives“ as part 7. instead of 6. since part 6.“ The potential applications of alleviating Al stress in crop production“ is given on lines 368-401, i.e., prior to part 7. Conclusions and perspectives.

Conclusions and perspectives, line 417: Modify the word form „explore“ to „exploration“ and remove the words „practical application values“ in the statement: „Therefore, investigation of functional mechanism of Al-SR genes, and exploration new methods to mitigating Al stress are formidable tasks to enhance crop grain yield.“ „These taska should be given priority considerations in future work.“

Final recommendation: Accept after a minor revision.

Author Response

  • The review manuscript entitled „Systematic investigation of aluminum stress-related genes and their critical roles in plants“ represents a valuable overview focused on aluminum stress-related genes in plants including transporters, transcription factors, signaling proteins including kinases and phosphatases, sugar metabolism-related proteins, ROS-related, and other processes related proteins. The major plant species where these proteins are discussed include Arabidopsis, rice, and maize. I can recommend the review manuscript for publication in IJMS due to the importance of the topic; however, I have some important comments on the present manuscript:

    Response: We thank you for your valuable comments, which have allowed us to improve the paper. The following changes have been made.

    1/ Figure 2: I think that Figure 2 providing just a schematic overview on cloned aluminum stress-related genes in plants and contains just information provided in Table 1 in a greater detail. Thus I think that Figure 2 is not necessary and should be removed.

    Response: Thank you for your friendly suggestion, we have removed Figure 2 in our manuscripts, and adjusted the following pictures of “Figure 3 to Figure 5” into “Figure 2 to Figure 4”.

     2/ Terminology:  The spelling of some important terms has to be correcetd, e.g., „ROS-related“ and „other processes“ instead of „Ros-relatedů and „other prosses“ both in Table 1 and the manuscript text.

    Response: Thank you for your friendly reminder, we have modified both “ROS-related” into “ROS metabolism” (lines 55, 81-82, 218, 237, 324, and Table 1) and “other prosses” into “other processes” (lines 55, 82, 239, Table 1, and Table S1).

    3/ Other formal comments on the text:

    Line 55: Correct the typing error in the term „other processes“ (not „other prosses“).

    Response: Thank you for your reminder, we have modified the “other prosses” into “other processes” in line 55.

    Line 89: Modify the statement as follows: „These results indicate that response to Al stress in plants may be underlied by participation of multiple organelles.“

    Response: Thank you for your meaningful suggestion, we have modified the “These results indicate that response to Al stress in plants may be underlied by participation of multiple organelles” into “These results indicate that the response to Al stress may take place in various organelles in plants.” in line 88-89.”

    Line 159: Add „a“ prior to the words „common role“, i.e., „…play a common role in Al stress…“

    Response: According to your suggestion, we have added “a” prior to the words “common role“ in line 142.

    Line 161: Correct the typing error in the term „Al stress“ (not „al stress“).

    Response: We have modified the “al stress” into “Al stress” in line 144.

    Line 179: Add „a“ prior to the words „core role“ in the statement: „Indicating that those transcription factors may play a core role in plants under Al stress.“

    Response: According to your suggestion, we have added “a” prior to the words “core role“ in line 162.

     Line 202: Use the words „enhanced susceptibility“ instead of „heightened susceptibility“.

    Response: According to your suggestion, we have used “enhanced susceptibility“ instead of “heightened susceptibility“ in line 185.

    Line 210: Add „an“ preceding the words „important role“ i.e., „…play an important role in Al accumulated in root cell wall…“

    Response: According to your suggestion, we have added “an” prior to the words “important role“ in line 193.

    Line 213: Correct the typing error in the chemical name „endo- trans-glucosylase-hydrolase“ (not „edotransglucosylase-hydrolase“).

    Response: According to your suggestion, we have modified the “edotransglucosylase-hydrolase” into “endo- trans-glucosylase-hydrolase” in line 196.”

    Line 234: Write „ROS-related proteins“ with capital letters in „ROS“ (not „Ros“).

    Response: According to your suggestion, we have modified the “Ros” into “ROS” in line 217.”

    Line 256: Correct the typing error in the heading „3.7. Other processes“ (not „Other prosses“).

    Response: According to your suggestion, we have modified the “Other prosses” into “Other processes” in line 239.”

    Line 402: Renumber the part „Conclusions and perspectives“ as part 7. instead of 6. since part 6.“ The potential applications of alleviating Al stress in crop production“ is given on lines 368-401, i.e., prior to part 7. Conclusions and perspectives.

    Response: According to your suggestion, we have modified the “Conclusions and perspectives“ as part 7.

    Conclusions and perspectives, line 417: Modify the word form „explore“ to „exploration“ and remove the words „practical application values“ in the statement: „Therefore, investigation of functional mechanism of Al-SR genes, and exploration new methods to mitigating Al stress are formidable tasks to enhance crop grain yield.“ „These taska should be given priority considerations in future work.“

    Response: According to your suggestion, we have modified “explore” to “exploration”, removed “practical application values” in line 414-417. And used “Therefore, investigation of functional mechanism of Al-SR genes, and exploration new methods to mitigating Al stress are formidable tasks to enhance crop grain yield. These tasks should be given priority considerations in future work.” instead of “Therefore, investigation of functional mechanism of Al-SR genes, practical application values, and explore new methods to mitigating Al stress are formidable tasks to enhance crop grain yield, should be given priority considerations in future work.”.

Reviewer 2 Report

Comments and Suggestions for Authors

This work is related to the Systematic Investigation of Aluminum Stress-related Genes and Their Critical Roles in plants. It is a topic that has been worked on a lot, yet this manuscript is important as a reference document because it adequately summarizes the genes related to aluminum stress.

Although there are numerous publications on this topic, this manuscript relates them in a very intelligent way that summarizes the topic in question very well.

As this is a review, no methodologies are shown.

1. I would like to comment that it would be important to comment on the genes that are involved in the beneficial effects of aluminum in some species Bojórquez-Quintal E, Escalante-Magaña C, Echevarría-Machado I and

Martínez-Estévez M (2017) Aluminum, a Friend or Foe of Higher Plants in Acid Soils. Front. Plant Sci 8:1767. doi:10.3389/fpls.2017.01767.

2.the figures have very poor resolution, and it is difficult to see them properly, I suggest that the resolution be improved.

 The references are adequate and extensive.

Author Response

  • This work is related to the Systematic Investigation of Aluminum Stress-related Genes and Their Critical Roles in plants. It is a topic that has been worked on a lot, yet this manuscript is important as a reference document because it adequately summarizes the genes related to aluminum stress.

    Although there are numerous publications on this topic, this manuscript relates them in a very intelligent way that summarizes the topic in question very well.

    As this is a review, no methodologies are shown.

    Response: Thank you very much for your positive comments. The corresponding modification has been made in accordance with your good suggestion.

    1. I would like to comment that it would be important to comment on the genes that are involved in the beneficial effects of aluminum in some species Bojórquez-Quintal E, Escalante-Magaña C, Echevarría-Machado I and

    Martínez-Estévez M (2017) Aluminum, a Friend or Foe of Higher Plants in Acid Soils. Front. Plant Sci 8:1767. doi:10.3389/fpls.2017.01767.

    Response: Thank you for the good suggestions. Al does play pleiotropic functions of beneficial or toxic effect to plants and other organisms. This result has an important implication to study the influence of Al and efficient use of Al resource. In our manuscript, we have added more research information “Previous studies have shown that Al have pleiotropic functions of beneficial or toxic effect to plants and other organisms, which are depending on factors such as, metal concentration, the chemical form of Al, growth conditions and plant species [157]. So, mitigating the Al stress even to efficient use of Al resource is very necessary agricultural production.” and added one reference of [157] “Bojórquez-Quintal, Emanuel; Escalante-Magaña, Camilo; Echevarría-Machado, Ileana; Martínez-Estévez, Manuel. Aluminum, a friend or foe of higher plants in acid soils. Front. Plant. Sci. 2017, 8, 1767. Doi:10.3389/fpls.2017.01767” in lines 364-3368 of part 6.

    2.the figures have very poor resolution, and it is difficult to see them properly, I suggest that the resolution be improved.

    Response: According to your good suggestion, we have improved the resolution of the figures.